# Nutrition in Necrotizing Enterocolitis and Following Intestinal Resection

**DOI:** 10.3390/nu12020520

**Published:** 2020-02-18

**Authors:** Jocelyn Ou, Cathleen M. Courtney, Allie E. Steinberger, Maria E. Tecos, Brad W. Warner

**Affiliations:** 1Department of Pediatrics, Division of Newborn Medicine, Washington University School of Medicine, St. Louis, MO 63110, USA; jocelyn.ou@wustl.edu; 2Department of Surgery, Division of Pediatric Surgery, Washington University School of Medicine, St. Louis, MO 63110, USA; c.courtney@wustl.edu (C.M.C.); allie.steinberger@wustl.edu (A.E.S.); metecos@wustl.edu (M.E.T.)

**Keywords:** necrotizing enterocolitis, prematurity, intestinal resection, short bowel syndrome, intestinal adaptation, microbiome, parenteral nutrition, hormones, breast milk

## Abstract

This review aims to discuss the role of nutrition and feeding practices in necrotizing enterocolitis (NEC), NEC prevention, and its complications, including surgical treatment. A thorough PubMed search was performed with a focus on meta-analyses and randomized controlled trials when available. There are several variables in nutrition and the feeding of preterm infants with the intention of preventing necrotizing enterocolitis (NEC). Starting feeds later rather than earlier, advancing feeds slowly and continuous feeds have not been shown to prevent NEC and breast milk remains the only effective prevention strategy. The lack of medical treatment options for NEC often leads to disease progression requiring surgical resection. Following resection, intestinal adaptation occurs, during which villi lengthen and crypts deepen to increase the functional capacity of remaining bowel. The effect of macronutrients on intestinal adaptation has been extensively studied in animal models. Clinically, the length and portion of intestine that is resected may lead to patients requiring parenteral nutrition, which is also reviewed here. There remain significant gaps in knowledge surrounding many of the nutritional aspects of NEC and more research is needed to determine optimal feeding approaches to prevent NEC, particularly in infants younger than 28 weeks and <1000 grams. Additional research is also needed to identify biomarkers reflecting intestinal recovery following NEC diagnosis individualize when feedings should be safely resumed for each patient.

## 1. Introduction

Necrotizing enterocolitis (NEC) remains one of the most devastating diagnoses in premature neonates. Although its incidence varies amongst different neonatal intensive care units, the mean prevalence of NEC is 7% in infants between 500–1500 grams and the disease has a high morbidity and mortality [1]. The exact pathophysiology of NEC is unknown, but the immature intestinal barrier and intestinal dysbiosis are two important factors that likely contribute to intestinal inflammation and injury seen in the disease [1,2]. Because of its nonspecific symptoms, NEC is difficult to diagnose. Currently, Bell’s staging, first introduced in 1978 by Bell et al. and modified by Kligeman and Walsh in 1986, is widely used to stratify disease severity and guide treatment (Figure 1). For Bell’s stage 1 (suspected, but not confirmed NEC) and Bell’s stage 2 (confirmed pneumatosis intestinalis with or without portal venous gas) [2], parenteral nutrition (PN) and broad-spectrum antibiotics are initiated, and enteral feeds are held for 7–14 days. Because the management of disease in these stages is non-operative, Bell’s stage 1 and Bell’s stage 2 are also known as “medical NEC.” If disease progresses despite holding feeds and starting antibiotics, surgery is required in Bell’s stage 3, which is characterized by hemodynamic instability in addition to severe thrombocytopenia, disseminated intravascular coagulopathy, and peritonitis (IIA) or pneumoperitoneum (IIB) [2,3]. Surgical NEC increases disease mortality from 3% to 30% [4]. Not infrequently, the length of bowel needed to be removed can be significant, resulting in short bowel syndrome. 

On a cellular level, intestinal adaptation occurs after massive bowel resection as a compensatory response by the remnant bowel wherein villi elongate, crypts deepen, and enterocyte proliferation is enhanced. Together, these changes function to increase the functional absorptive capacity per unit length of the remnant bowel [5]. This review aims to summarize the role of nutrition in NEC, including its prevention, complications, and sequelae of surgical treatment. A thorough PubMed search was performed using search terms that included “preterm enteral feeding”, “early enteral feeding”, “feeding necrotizing enterocolitis”, “intestinal adaptation”, “intestinal adaptation macronutrients” and “parenteral nutrition necrotizing enterocolitis.” Meta-analyses and randomized controlled trials were reviewed on these topics when available; otherwise, pre-clinical animal trials were reviewed. We included studies pertaining to nutrition in NEC, specifically those examining feeding comparisons in which NEC was a primary or secondary outcome. Reports not focused on NEC as a primary or secondary outcome or those discussing NEC without a clear definition were excluded.

## 2. NEC Prevention 

### 2.1. Delivery of Feeds

#### 2.1.1. Initiation of Feeds

Historically, it was thought that delaying enteral feeds would decrease the incidence of NEC. However, a 2013 Cochrane review found that there was no increased incidence of NEC when beginning trophic feeds early (within 96 hours of birth) and continuing them for a week compared to fasting and starting feeds at 7 or more days of life in very preterm (<32 weeks) or very low birthweight (<1500 grams) infants [6]. Starting enteral nutrition early also was not protective against NEC in this population. Initiating enteral feeds early appears to be safe for preterm and very low birthweight infants and beginning feeds at a higher volume may also be beneficial. A 2019 meta-analysis conducted by Alshaikh, et al. compared the safety of starting total enteral feeds at 80 ml/kg/d to starting enteral feeds at the conventional volume of 20 ml/kg/d. There was no difference in the incidence of NEC or feeding intolerance when starting early total enteral feeds, with the added benefit of decreased late-onset sepsis and decreased length of hospital stay by 1.3 days [7]. However, the conclusions that can be drawn from this meta-analysis for infants <1000 g and <28 weeks are limited, as the studies included in the analysis included infants between 28–36 weeks’ gestational age and between 1000–1500 g. In a 2019 randomized controlled trial, Nangia, et al. compared starting very low birthweight infants between 28–34 weeks’ gestational age on total enteral feeds (80 ml/kg/day) on the first day of life to starting infants at the conventional enteral feeding volume (20 ml/kg/day) supplemented with intravenous fluids. The study found no difference in the incidence of NEC between the two groups, with 1.1% in the early total enteral feeding group compared to 5.8% in the conventional enteral feeding group (*p* = 0.12). However, infants in the early total enteral feeding group reached goal feeds on average of 3.6 days sooner. This group also had fewer complications such as sepsis or feeding intolerance, and ultimately had shorter lengths of stay [8].

#### 2.1.2. Feeding Advancement

Once feeds are successfully initiated and tolerated, the next consideration is the rate of feed advancement. Although there is significant variation in advancement protocols amongst different neonatal intensive care units, feeds are typically increased by 15–35 ml/kg each day, depending on infant size. Dorling, et al. conducted a randomized controlled trial comparing slow (18 ml/kg/day) and rapid (30 ml/kg/day) feed advancement that showed no significant difference in survival without moderate or severe neurologic deficits at 24 months in very preterm (<32 weeks) and very low birth weight infants [9]. Rapid advancement of feeds also did not increase the incidence of NEC when compared to slow advancement. Advancing feeds more rapidly and thus allowing infants to reach full feeds sooner may lead to increased caloric intake and better growth, as well as decreased duration of parenteral nutrition. 

#### 2.1.3. Bolus and Continuous Feeding

Bolus feeding has the advantage of gut stimulation, which promotes normal functioning and tissue maturation. Conversely, continuous feeding provides an opportunity for slow and steady nutrient introduction, which may allow for better tolerance and absorption in the setting of less distension and diarrhea [10,11]. In a recent meta-analysis, Wang, et al. found that although there was no difference in growth parameters or length of hospitalization, bolus-fed preterm (<37 weeks’ gestational age), low birthweight (<2500 grams) infants reached feeds sooner (mean difference 0.98 days) with a similar incidence of NEC compared to infants receiving continuous feeds [12]. This meta-analysis includes infants up to 2500 grams, but found no differences in subgroup analysis of infants with birthweight <1000 grams and >1000 grams.

Randomized controlled trials have disproven previous observational data that delaying the initiation of feeds, starting at a smaller volume, and advancing feeds slowly may decrease the incidence of NEC. Evidence remains limited in extremely preterm and extremely low birthweight infants; a feasible approach to feeding preterm infants may be initiating feeds as soon as an infant is clinically stable and advancing by 30 ml/kg/day as tolerated. For very low birthweight infants, starting feeds within 96 hours of birth and advancing at 30 ml/kg/day have both been shown to be safe and allow infants to reach full feeds sooner. However, despite decreasing the number of days infants require parenteral nutrition, advancing feeds faster does not decrease the incidence of late-onset sepsis and in general, the benefit of reaching full feeds faster may be limited. The most beneficial approach may be for each neonatal intensive care unit to standardize their feeding protocols and ensure that are consistently followed.

### 2.2. Composition of Feeds

#### 2.2.1. Osmolality

Human breast milk has an osmolality of around 300 mOsm/l, whereas commercially available enteral formulas have osmolalities of less than 450 mOsm/l [13]. In order to meet a preterm infant’s nutritional and growth requirements, both breast milk and infant formulas require caloric fortification and supplements, thereby increasing osmolarity. Multi-nutrient fortification adds protein, vitamins, and other minerals and increases the osmolality of breast milk to 400 mOsm/l [13]. Historically, administration of hyperosmolar formula was thought to be associated with an increased risk for the development of necrotizing enterocolitis (NEC). This was based on a handful of small-scale studies in the 1970s, all of which failed to provide a durable mechanism of mucosal injury [14,15]. More recently, Miyake, et al. looked at hyperosmolar enteral formula compared to diluted formula in a mouse model of NEC. They found that the inflammatory response, mucosal injury, and incidence of NEC was the same in both experimental groups [16]. In other animal studies, the only reported adverse outcome associated with hyperosmolar feeds was delayed gastric emptying [13]. Lastly, in humans, a 2016 Cochrane review concluded that there is weak evidence showing that nutrient fortification does not increase the incidence of NEC in preterm infants. It does increase in-hospital growth rate (weight 1.81 g/kg/day, length 0.12 cm/week, head circumference 0.08 cm/week), but does not seem to improve long-term growth and development [17]. Because in-hospital growth rates are improved and the incidence of NEC is not increased with hyperosmolar feeds, the benefit of additional nutrients and other supplements warrants fortification of human breast milk. The data on the effect of fortification on neurodevelopment and growth beyond infancy is very limited and needs to be studied further.

#### 2.2.2. Breast Milk 

Human milk is the only modifiable risk factor that has been consistently shown to protect against the development of NEC [18,19]. Since the 1990s, the incidence of NEC has been described as 6–10 times higher in exclusively formula-fed infants compared to exclusively breastfed infants [20]. The specific mechanisms by which breast milk is protective continue to be studied. However, several non-nutrient components have been found to contribute to the immune functions of the gastrointestinal tract and augment mucosal integrity [21,22]. These include secretory IgA, growth hormones (epidermal growth factor, insulin, and insulin-like growth factor), polyunsaturated fatty acids, and oligosaccharides. A 2019 study found that not only is an infant’s IgA largely derived from maternal milk in the first month of life, but also that infants with NEC have larger proportions of IgA-unbound bacteria compared to age-matched controls. In the same study, Gopalakrishna, et al. used a murine model and concluded that pups reared by IgA-deficient mothers were not protected from NEC [23]. It has also been hypothesized that the beneficial effects of human milk relate to how diet affects gut microbiota and the developing immune system. Human breast milk contains oligosaccharides known to stimulate “healthy” bacteria and in a murine model, has been shown to downregulate bacterial related inflammatory signaling pathways [24].

#### 2.2.3. Donor Breast Milk

Although mother’s own milk is preferred for preterm and low birth weight infants, infants often need to be supplemented with donor breast milk or formula when maternal supply is inadequate. Donor milk has also been shown to have a protective effect on NEC incidence when compared to cow’s milk and other formulas, with a 79% reduced risk [25,26,27,28]. A 2019 Cochrane Database review of 12 trials found that although formula-fed or supplemented preterm and low birth weight infants did have increased growth compared to those fed with donor breast milk, they also exhibited a higher risk of NEC (typical risk ratio 1.87) [29]. 

#### 2.2.4. Cow’s Milk Formula

Prior literature has established a higher incidence of NEC when cow’s milk formula is used instead of mother’s own milk [25,30]. In addition to the protective factors that breast milk contains, it’s been hypothesized that the intestinal reaction to cow’s milk proteins could also contribute to disease pathogenesis. In small cohorts of infants with NEC, a group has found an increase in cytokine response (interferon-γ, IL-4, and IL-5) to cow’s milk proteins beta-lactoglobulin and casein [31]. Interestingly, bovine milk-derived exosomes have been shown to combat experimentally induced NEC by stimulating goblet cells and mitigating decreases in mucin 2 (MUC2) and glucose regulated protein 94 (GRP94). Isolation and administration of such exosomes could be useful for infants at high risk for NEC for whom breast milk cannot be obtained [32].

## 3. Medical NEC

Symptoms seen in the early stages of NEC may mimic feeding intolerance or other abdominal pathologies. The modified Bell’s staging criteria include neutropenia, thrombocytopenia, coagulation factors, and metabolic acidosis as laboratory markers that can aid clinicians in diagnosing more advanced NEC [3]. However, these laboratory values are non-specific and are less likely to be reliable markers for disease in early stages or to predict intestinal recovery and safety to restart feeds. In addition to antibiotics, current nutritional management for NEC includes stopping feeds and starting parenteral nutrition.

### 3.1. NPO Duration

Patients’ *nil per os* (NPO) status is largely driven by clinical assessment. Decreased apneic and bradycardic events in conjunction with laboratory values including blood gas, white count, and thrombocytopenia, as well as abdominal imaging without the appearance of portal venous gas or pneumatosis intestinalis are indications of improving clinical status [33]. Despite apparent improvement in clinical status, clinicians may hesitate to restart feeds after an NPO period, as objective evidence reflecting the optimal time to begin feeding is lacking. A meta-analysis conducted by Hock, et al. found no significant difference in adverse outcomes in patients given early (within 5 days of NEC diagnosis) and late (>5 days after NEC diagnosis) feeds [34]. Bonhorst, et al. utilized ultrasonography and compared outcomes following restarting feeding after 3 consecutive days without portal venous gas to restarting feeding after 10 days without portal venous gas. Earlier feeds were associated with fewer complications, shorter antibiotic courses, quicker progression to goal feeds, and shorter length of stay [35]. 

In addition to using imaging as an objective measure for readiness and safety to restart feeds, specific laboratory values and biomarkers would be useful. In a 2019 prospective observational cohort study, Kuik, et al. measured the regional intestinal oxygen saturation (r_int_SO_2_) by near-infrared spectroscopy (NIRS) and intestinal fatty acid binding protein (I-FABP_u_) in the urine of 27 preterm infants. The study found that when measured after the first re-feed, these markers were predictive of post-NEC stricture, though not of recurrent NEC [36]. Additionally, a recent study on infants between 24–40 weeks postmenstrual age found high intestinal alkaline phosphatase (IAP) in stool and low IAP enzyme activity in patients with NEC compared to those without disease; IAP also was a useful biomarker for disease severity [37]. Clinicians should attempt to minimize NPO time and begin refeeding patients as soon as clinical improvement is determined by vital sign stability and abdominal examination, as well as resolving thrombocytopenia and abdominal radiography or ultrasonography. Identifying biomarkers such as IAP that reflect a patient’s disease severity and intestinal recovery could be useful in individualizing NPO duration to minimize complications associated with prolonged NPO status.

### 3.2. Parenteral Nutrition

PN is initiated in patients who are made NPO following NEC diagnosis. It is comprised of carbohydrates, amino acids, lipids, electrolytes, minerals, and vitamins administered intravenously to allow for bowel rest. PN should be started early with adequate protein (3.5–4 g/kg/day) to maintain a positive nitrogen balance, improve weight gain, and to allow repair of injured tissue [1,38,39,40]. However, it has been shown that supplemental PN at NEC onset does not appear to significantly improve outcomes, with no decrease in the rate of surgical intervention or in-hospital mortality [41]. PN is discontinued once enteral feedings approach goal volumes [42]. 

## 4. Surgical NEC

In cases of NEC refractory to medical management or NEC leading to intestinal perforation, surgery is indicated (i.e., “surgical NEC”). A complication of NEC following extensive intestinal resection is short bowel syndrome (SBS) and subsequent intestinal failure (IF) wherein the small bowel is unable to adequately absorb fluids, electrolytes, and nutrients required to support growth and development [43]. Nutrition therefore must be provided through parenteral nutrition. The key compensatory process involved in reaching enteral autonomy is intestinal adaptation. Adaptation is characterized by structural and functional changes that compensate for the loss of intestinal mucosal surface area [44] and involves an increase in villus height and crypt depth, myocyte and enterocyte proliferation, a decrease in enterocyte apoptosis, and elongation and dilatation of the remnant small bowel [45]. Therefore, post-operative nutrition strategies focused on enhancing the intestinal adaptive response remain a cornerstone of treatment. Factors known to play important roles in adaptation and enteral autonomy include length of remnant bowel, specific macronutrients, and the composition of PN.

### 4.1. Enteral Feeding

While the optimal enteral formulation for pediatric SBS is still unknown, the data consistently supports the benefit of breast milk in intestinal adaptation [46]. In addition to growth factors and immunoglobulins, breast milk contains key oligosaccharides that act in a prebiotic manner to stimulate enterocyte proliferation and positively regulate the intestinal microbiome [47]. The most abundant of these is 2’-fucosyllactose (2’-FL). A few preclinical studies have investigated the effects of 2’-FL enteral supplementation on various intestinal inflammatory diseases. Mezoff, et al. demonstrated that 2’-FL augments intestinal adaptation after ileocecal resection by optimizing energy processing by the gut microbiome [48]. Another group showed that 2’-FL significantly decreased the severity of colitis in interleukin-10 null mice through enhanced epithelial integrity and expansion of a positive gut microbial environment [47]. 

### 4.2. Anatomical Considerations

#### 4.2.1. Intestinal Length 

Following surgical NEC, remnant length and anatomy become major determinants of disease severity [49]. It is well demonstrated that residual intestinal length is inversely proportional both to duration of PN and mortality [50,51,52]. While there is no definitive threshold, data suggests that greater than 40 cm of remnant small bowel length (SBL) in the presence or absence of an ileocecal valve (ICV) is associated with improved outcomes [50]. The effect of an intact ICV is somewhat controversial and likely a surrogate for the presence of colonic mucosa. This may be more important in patients with less than 15 cm [53]. Quiros-Tejeira, et al. showed that both survival and enteral adaptation were increased when more than 38 cm of small bowel length remained [50]. Lastly, Goulet, et al. analyzed 87 SBS children based on PN weaning and reported that all patients in the PN-dependent group had less than 40 cm of SBL and/or absent ICV. Conversely, patients with persistent enteral independence had SBL of 57 +/− 19 cm [51]. Given the rapid intestinal elongation that normally occurs in late gestation, studies have recommended using the percentage of expected length as opposed to absolute remnant length in neonates. By this metric, Spencer, et al. found that greater than 10% age-adjusted remnant bowel length was highly predictive of both survival and enteral autonomy [52].

#### 4.2.2. Segment Functionality

Given the segmental functionality of the gastrointestinal tract, the site of bowel resection has a substantial impact on the need for long-term nutritional support [54]. The three most common resection patterns in SBS are jejunoileal anastomosis, jejunocolic anastomosis, and jejunostomy. These anatomical permutations are associated with a predictable range of outcomes based on digestive and absorptive capacity. 

Patients with jejunoileal anastomoses (mostly jejenum removed) have the highest likelihood of achieving enteral autonomy. This proximal resection spares the ileum, which has the greatest capacity for structural and functional adaptation [55]. In addition, the presence of the ileocecal valve and colonic continuity may mitigate intestinal transit time and excessive fluid losses [54]. Despite the intestinal adaptive capacity of patients with jejunoileal anastamoses, this population still experiences gastric hypersecretion secondary to loss of regulatory humoral action (cholecystokinin, secretin, vasoactive intestinal peptide, and serotonin) in the jejunum. This can transiently affect intestinal motility and increase gastric emptying and acid output. Administration of H_2_ antagonists and proton pump inhibitors may be helpful [56].

Patients with jejunocolic anastomoses (mostly ileum removed) are often more difficult to manage, as the jejunum lacks the robust adaptive capacity of the distal small bowel [56]. Decreased water absorption along the proximal remnant length overwhelms the compensatory abilities of the colon, leading to fluid and electrolyte losses through diarrhea [54]. Furthermore, the ileum is the primary site of vitamin B12 and bile salt absorption. Consequent disruptions of the enterohepatic circulation result in fat malabsorption, steatorrhea, marked vitamin deficiencies, and renal oxalate stones [56]. Lastly, ileal resection can impact local hormonal control of gut motility through dysregulation of enteroglucagon and peptide YY [54]. As discussed above, loss of the ICV may be a negative predictor of long-term enteral autonomy. The ICV may play a role in the prevention of colonic bacterial migration into a small bowel environment that is vulnerable to bacterial overgrowth [57]. Ileocolic resections will result in variable PN dependence, which is higher when less than 60 cm of proximal SBL remains [54].

End jejunostomy patients have the most severe malabsorptive phenotype and the highest likelihood of requiring long-term parenteral support [55]. In addition to the specific issues encountered with ileal resections, this population also lacks any of the absorptive, digestive and energy-salvaging compensation afforded by colonic continuity [54]. Accelerated rates of gastric emptying and intestinal transit due to changes in the intestinal hormonal milieu further minimize nutrient interaction with jejunal luminal mucosa. Net losses of fluid and electrolytes from high enterostomy output often exceed patient intake, necessitating supplementation with PN and intravenous fluid administration [56]. These patients must be carefully monitored for dehydration, metabolic disturbances and nutrient deficiencies. 

### 4.3. Ostomy Replacement

Fluid and electrolyte losses are significant problems in the pediatric SBS population and require diligent monitoring and repletion. This is especially true for children with small bowel ostomies. The degree of malabsorption, dehydration and metabolic disturbances are commensurate to the length of small intestine remaining and the site of resection [58]. While an adaptive compensatory response is seen in patients with ileostomies, there is little evidence of structural or functional adaptation in those with jejunostomies. Despite optimized nutritional management and fluid balance, these patients are likely to require prolonged PN [58]. Furthermore, if less than 75 cm of small bowel remains in the presence of a jejunostomy, the ability to wean from parenteral nutritional or saline support is significantly impaired [58]. Patients with SBS and enterostomies tend to lose considerable amounts of sodium in stool causing secondary hyperaldosteronemia and significant potassium losses in the urine [59]. This often requires separate parenteral saline repletion in addition to the sodium provided from PN in amounts up to 8–10 mEq/kg/day [59]. Ostomy output and electrolytes should be closely observed to maintain hydration with urine output of at least 1–2 ml/kg/day and urine sodium >30 mEq/L [59]. 

High ostomy output is generally defined as greater than 40 ml/kg per 24 hours, with the severity of losses highly dependent on the length and site of remaining bowel. Provision of adequate fluids to prevent and treat dehydration is tantamount in this population, as the risk of hypotension and pre-renal failure are high [58]. Fluid needs are typically delivered through a combination of PN and EN, but may require supplemental intravenous fluids in cases of excessive loss [49]. 

### 4.4. Macronutrients

#### 4.4.1. Fat

Several preclinical studies have shown that lipids in particular are associated with an enhanced adaptation response. Rats fed high fat diets (HFD) had significantly increased bowel weight and villus height post-resection when compared to those fed standard chow [60]. Choi, et al. randomized mice to low (12% kcal fat), medium (44% kcal fat) and high (71% kcal fat) fat diets after 50% proximal small bowel resection (SBR) and demonstrated that increased enteral fat concentration (HFD) optimally prevented postoperative catabolic responses and increased lean mass after SBR [61]. Conversely, in another rat model, low fat diets, despite comparable caloric intake, negatively impacted adaptation as evidenced by decreased body weight, reduced expression of fat transporters and attenuated villus height and enterocyte proliferation [62]. 

Moreover, the specific kind of enteral fat appears to play an important role in intestinal adaptation. In rats, long-chain fatty acids (LCFA) are superior to medium-chain fatty acids (MCFA) in augmenting both the structural and functional intestinal response following SBR [63]. While most studies have focused on polyunsaturated LCFA (LCPUFAs) such as menhaden oil, the relative benefit compared to saturated FAs is still debated. Menhaden oil is an excellent source of the omega-3 fatty acids eicosapentaenoic acid (EPA) and docosahexanoic acid (DHA) [64]. EPA and DHA are not only precursors of anti-inflammatory prostaglandins and associated with improved cardiovascular profiles, but they have also been shown to enhance intestinal adaptation after massive small bowel resection. Kollman, et al. demonstrated that resected rats fed LCPUFA-enhanced diets demonstrated significantly increased intestinal mucosal mass in a dose-dependent manner [65]. Another study found that in a mouse model, menhaden oil (versus saturated and monounsaturated fats) resulted in the highest percent of lean mass and greatest weight retention after SBR, though adaptation was indistinguishable across diets [66]. The benefit of LCFAs is attributed to its anti-inflammatory metabolite (prostaglandins), as inhibition with aspirin, a cyclo-oxygenase inhibitor, reduces the predicted intestinal adaptive response [67]. Although LCFAs have the greatest trophic yield, their absorption can be suboptimal in patients with extensive distal resections due to compromised enterohepatic circulation. While MCFAs are more water soluble, they have not been shown to have a robust effect on adaptation in mice and have significant osmotic sequelae which can exacerbate diarrhea and fluid losses [68]. Most of what is known about the effect of fats on adaptation is from preclinical animal models, but a high fat diet, specifically with LCFA, have been shown to support adaptation in these models and could potentially increase intestinal adaptation in patients following resection as well. 

#### 4.4.2. Protein

Most of the literature surrounding the protein composition of enteral nutrition is focused on absorption rather than adaptation. Elemental (fully digested) or semi-elemental (partially digested) enteral formulas have historically been preferred in infants with SBS in an effort to maximize absorption in the remnant bowel. For a subset of patients, losses can occur both in the bowel effluent and through loss of protein exudate. This double hit is akin to a protein-losing enteropathy, necessitating increased protein requirements for adequate growth. The extent to which children with persistent malabsorption and intolerance may benefit from a hydrolyzed formula is not known. A small study of four children with SBS found that after initiating a hydrolyzed formula, subjects that previously had persistent feeding intolerance were able to be weaned off parenteral nutrition within 15 months [69]. A possible explanation for improved tolerance on hydrolyzed formula could be non-IgE mediated cow’s milk protein sensitization seen in infants with NEC [31]. However, it is difficult to draw conclusions from the small population that was observed. Additionally, it has been shown that 70–90% of protein absorption ability is retained after massive intestinal resection in human neonates [70]. It was previously theorized that the lack of MCTs and lactose in extensively hydrolyzed formulas may lead to easier digestion in patients with SBS. However, in a randomized crossover trial comparing protein hydrosylate formula to standard formulas in children with SBS, Ksiazyk et al. found no differences in intestinal permeability, energy expenditure, or nitrogen balance [46].

Providing adequate amino acids after intestinal resection is important. Glutamine serves as the primary fuel substrate for intestinal cells, promoting enterocyte proliferation and protein synthesis [71]. In preclinical rat models, there is a marked increase in glutamine and total amino acid uptake in the early adaptive phases following SBR. Unfortunately, supplementing enteral nutrition with glutamine or arginine after massive intestinal loss in humans has failed to improve adaptive responses and thus remains controversial [71,72]. Additionally, recent data suggests that complex nutrients promote greater intestinal adaptation. In this “functional workload” hypothesis, the remnant bowel meets the digestive demand of the nutrients encountered and there is thus a more robust compensatory response when infants are fed a non-hydrolyzed formula [57].

Ultimately, optimal protein intake from enteral nutrition should take into consideration remnant bowel length, absorptive capacity, and feeding tolerance. The goal is to achieve a positive nitrogen balance through improved nitrogen absorption. The data on the impact of formula protein content and composition on intestinal adaptation is sparse and the variation amongst formulas makes comparison of studies difficult. Although there is no robust evidence that elemental formula is superior to non-hydrolyzed formula, there is data showing that patients with SBS may tolerate it better and it is commonly used in the pediatric SBS population.

#### 4.4.3. Oligosaccharides

After intestinal resection, the bowel undergoes significant functional adaptation as evidenced in a rat model by increased densities of both key digestive enzymes and glucose transporters [64]. Excessive administration of simple carbohydrates should be avoided given their considerable osmotic effects.

Energy can be derived from complex carbohydrates and soluble fibers processed in the colon. These undigested macromolecules are metabolized by colonic bacteria to produce short chain fatty acids (SCFAs) such as butyrate [67]. Butyrate is the primary fuel substrate for colonocytes and has been shown to play an important role in intestinal adaptation in both rats [73] and piglets [48]. In neonatal piglets that underwent 80% distal SBR, butyrate supplementation markedly increased the structural and functional indices of intestinal adaptation in both the acute and chronic phases [48]. Similar findings were recapitulated using a rodent SBS model. DNA, RNA and protein content per unit mucosal weight all increased post-resection in fiber- and butyrate-supplemented diet compared to controls [73]. In humans, these benefits are mitigated by the length of residual colon and colonic continuity. Furthermore, simple carbohydrates in excess also have significant osmotic influence that may exacerbate diarrhea and extraneous losses [57]. Preferably, carbohydrates should comprise no more than 40% of the total caloric provision [57].

### 4.5. Parenteral Nutrition 

Surgical NEC typically delays the time until enteral autonomy and prolonged PN use (>21 days) may be required. Allin, et al. demonstrated that the need for PN support at 28 days post-decision to intervene surgically is associated with increased one-year mortality [74]. In clinical practice, intestinal insufficiency may be indirectly measured by the level of PN required for normal or catch up growth [75]. Patients with less remaining bowel require more PN and a residual length of 15–40 cm is associated with PN weaning [50,51,76,77,78]. The primary metabolic complication associated with PN is intestinal failure associated liver disease (IFALD), which is characterized by direct hyperbilirubinemia, elevated transaminases, and liver synthetic dysfunction [53]. Some modifications to PN can be made to reduce the risk for liver injury, such as not overfeeding and cycling infusions [42]. Improvement of cholestasis also depends on maintaining an appropriate protein-to-energy ratio in PN [79]. However, the most heavily studied factor implicated in PN-associated liver disease is intravenous lipid emulsions (ILE).

#### PN Lipid Source

ILEs are a crucial component of PN, as they are a source of essential fatty acids and non-protein calories. Several factors should be taken into consideration when choosing an ILE for parenteral use: the content of essential fatty acids (FAs), the ratio of polyunsaturated fatty acids omega-6 to omega-3, the quantity of α-tocopherol, and phytosterols. Monitoring FA profiles of children with IF is critical to their nutrition management.

Soybean-based (SO) lipid emulsions were previously considered the standard of care for providing fatty acids to children with intestinal failure. However, SO contains a 7:1 ratio of omega-6: omega-3, whereas the optimal ratio is 4:1 to minimize the production of inflammatory mediators [80,81]. It also has a high concentration of phytosterols, which have been associated with hepatic inflammation and cholestasis [82,83]. The SO factor, stigmasterol, has also been shown in a murine model to promote cholestasis, liver injury, and liver macrophage activation [84]. In 2012, Teitelbaum and colleagues described a significant reduction in cholestasis in a cohort of pediatric IF whose SO lipid dose was restricted to 1 g/kg/day compared to the historical dose of 3 g/kg/day [85]. Subsequent studies demonstrated that this lipid reduction strategy does not decrease the incidence of IFALD, but may slow its progression [86].

In 2018, the United States Food and Drug Administration approved a fish-oil (FO)-based lipid emulsion (Omegaven^®)^ for the treatment of pediatric intestinal failure associated liver disease (IFALD). Unlike SO-based lipid emulsion, FO is composed primarily of anti-inflammatory omega-3 FA (docosahexaenoic and eicosapentaenoic acids) and contains a small amount of the essential FA (linoleic and alpha-linolenic acids) [87]. FO-based lipids are rich in α-tocopherol, which scavenges free radicals from peroxidized lipids to prevent propagation of oxidative lipid damage [88]. IV FO treatment results in a biochemical reversal of cholestasis and is associated with reduction in plasma phytosterols, cytokines, and bile acids. However, despite biochemical and histologic improvement in cholestasis, there is persistent significant liver fibrosis on histology [89,90]. There is also concern that because FO provides fewer essential omega-6 FAs than that recommended in children, it could cause essential fatty acid deficiency (EFAD). However, Calkins, et al. found in a cohort of PN-dependent children, switching from SO to FO for six months led to a decrease in essential FA concentrations, but no evidence of EFAD [91]. These findings were supported in a long term study conducted over three years by Puder, et al. [92]. Newer preparations such as Smoflipid^®^ (Fresenius-Kabi, Uppsala, Sweden) combine soybean oil (30%), coconut oil (30%), olive oil (25%) and fish oil (15%) and have proven to be of benefit in patients with IFALD. Randomized controlled trials in preterm infants have shown that Smoflipid^®^ emulsion increases the content of eicosapentaenoic acid (EPA) and docosahexaenoic acids [93]. Muhammed, et al. reported rapid and marked improvement in biochemical liver function tests in children with cholestatic jaundice after switching from a SO-based ILE to Smoflipid^®^ [94]. Smoflipid^®^ has a positive impact on liver enzymes due low phytosterol and high vitamin E content; in addition, its use leads to a decrease in lipid peroxidation and an improvement on the *ω*-3:*ω*-6 PUFA ratio, producing a less proinflammatory profile [95]. 

## 5. Conclusions

Providing infants breast milk has been the mainstay of nutritional therapy in NEC prevention and is also beneficial for infants following surgery in stage III NEC [19,20,46]. Unfortunately, there have been no feeding strategies proven to prevent NEC, such as initiating feeds later, advancing feeds more slowly, or bolus versus continuous feeds; however, it is safe to start feeds within 96 hours of birth, advance more rapidly, and bolus feed [7,9,12]. Because there is great variability in individual feeding practices, it is important that each NICU has a standardized protocol to approaching feeds in order to ensure appropriate nutrition and minimize complications. Additional studies focusing on more premature and smaller infants should be conducted, as most studies that are currently available are limited to infants >1000 g and between 28–32 weeks. Younger and smaller infants may respond differently than older infants to alternate feeding approaches. Additionally, identifying more specific biomarkers for NEC severity and intestinal recovery is necessary to provide appropriate treatment and assist clinicians in determining intestinal recovery after disease. 

Finally, more diet studies on the effect of macronutrients on recovery after surgical NEC are required. The majority of current data on intestinal adaptation shows the benefit of a high fat diet but is limited to animal studies [61]. Using hydrolyzed formula in patients with SBS is common but has only been studied in a small population and lacks robust evidence [69]. Since parenteral nutrition is often required following resection, it is important to understand its complications. Omegaven^®^ and Smoflipid^®^ both are less likely to lead to cholestasis and IFALD without causing essential fatty acid deficiency and may be more beneficial as a fat source than the traditionally used intralipids [92,95]. 

## Figures and Tables

**Figure 1 nutrients-12-00520-f001:**
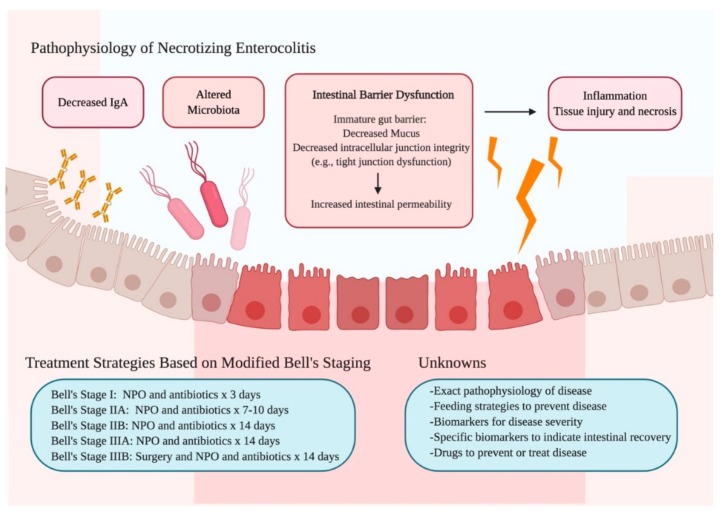
Summary of the Pathophysiology, Treatment Strategies, and Unknowns of Necrotizing Enterocolitis. The pathophysiology of NEC is multi-faceted, involving intestinal barrier dysfunction, decreased IgA, and altered microbiota. Current treatment strategies include stopping feeds and starting antibiotics based on disease severity, as classified by Bell’s staging. Much remains unknown about disease prevention, diagnosis, and treatment. Figure created with Biorender.com. Abbreviations: Immunoglobulin A (IgA), NEC (Necrotizing enterocolitis), NPO (*nil per os*).

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
