# Peer review of "Nutrition in Necrotizing Enterocolitis and Following Intestinal Resection"

_nutrients, 2020, doi:10.3390/nu12020520_

Round 1
Reviewer 1 Report
This narrative review discusses several aspects of the prevention and management of NEC and its complications such as short bowel syndrome following resection of necrotic bowel.
It should be thouroughly revised and almost totally rewritten.
The abstract should start with a clear research question regarding the aim of the review, and include some information on the method used (search strategy, criteria for inclusion / exclusion of references), followed by the most important results and a conclusion.
The introduction also should lead to a clear and focused research question. There are no methods described, this should be added: what search strategy has been used, which criteria have been used for inclusion/exclusion of references. The results need to be presented more clearly (see suggestions inserted as remarks in the paper). As the paper is presented now, there is no conclusion, there is no indication of what is known on the topic and which research questions remain open. The reader does not become “wiser” regarding how to feed premature patients following NEC with or without bowel resection. That breastfeeding is associated with a lower risk for NEC is generally known.
In general, the references are diverse, including systematic reviews such as cochrane reviews together with animal work. When the authors describe the findings they often do not mention it concerns animal work (for example ref 71 which is cited several times, when reading the text of the paper, it is not at all clear this reference concerns a mice model). I propose they either include animal work, but describe the results clearly separated from human data, or do not include animal work at all. When discussing results of studies, they should give an indication of the strength and clinical relevance of the findings.
While the title suggests the review is focused on nutrition, the authors also include a section on hormones and on the microbiome. It is not clear why. The should either broaden the scope of the review to management of NEC and its complications in general or limit to the possible role of nutrition in prevention and management of NEC and its complications (given the scope of the journal, I suggest the latter)
More specific remarks can be found in the annotated file of the submission (see attachment).

Reviewer 2 Report
General comments
The manuscript provides a solid description of several alimentary and non-alimentary factors affecting the development and post-operative outcome of NEC. However, the manuscript could be improved by addressing the points below:
The authors might consider including a workflow scheme summarizing the current knowledge of pathophysiology, current treatment options (incl. first-line treatment approaches), as well as the ‘unknowns’. The manuscript is lacking a conclusion, future outlook, a short summary or a ‘take-home message’ – what are the key points that need to be further improved? Is there a need for new biomarkers & earlier diagnosis? Should formula-feeding be largely discouraged in premature infants? Are breast milk donor programs available/useful?
Minor issues
The introduction should include the definition, current diagnostic criteria and up-to-date management recommendations of NEC. ‘Medical NEC’? – could this be replaced by ‘Medical management of NEC’? Or does this mean pharmacological rather than medical? Line 165 onwards – please provide the reference and a short description of Bell’s stages. Line 198 – please see point 2.Author Response
Please see attachment.

Round 2
Reviewer 1 Report
The manuscript has been greatly improved, and is now clearly focused, and contains a conclusion.
I still miss a clear method section (which search terms have been used, in- and exclusion criteria, …)
Remarks have been inserted in the text (see attachment)
